# *AOX* Affects the Synthesis of Polysaccharides by Regulating the Reactive Oxygen Species in *Ganoderma lucidum*

**DOI:** 10.3390/foods14050826

**Published:** 2025-02-27

**Authors:** Ruiying Zhu, Lele Zhang, Longxi Wu, Jingshuo Liu, Jie Zhang, Jian Li, Kejing Song, Peipei Han

**Affiliations:** Key Laboratory of Industrial Fermentation Microbiology, Ministry of Education, College of Biotechnology, State Key Laboratory of Food Nutrition and Safety, Tianjin University of Science & Technology, Tianjin 300457, China; zry@tust.edu.cn (R.Z.); lelezhang798@163.com (L.Z.); well20220320@163.com (L.W.); 15332130298@163.com (J.L.); zj117749@163.com (J.Z.); lov_lj@mail.tust.edu.cn (J.L.); kjsong@tust.edu.cn (K.S.)

**Keywords:** *G. lucidum*, alternative oxidase, reactive oxygen species, polysaccharides

## Abstract

Alternative oxidase (AOX) is a terminal oxidase in the mitochondrial electron transport chain that does not contribute to the generation of ATP. It plays a critical role in maintaining the balance between reactive oxygen species (ROS) production and intracellular redox homeostasis within the mitochondria. In the study, overexpression and knockdown approaches were employed to investigate the function of *AOX*. AOX-silenced strains (AOXi3 and AOXi25) and AOX-overexpressed strains (OE-AOX2 and OE-AOX21) were constructed. The ROS level and transcription level of the antioxidant-system-related genes, including phosphoglucomutase (*pgm*) and phosphomannose isomerase (*pmi*), were differentially upregulated in silenced strains, whereas the opposite effect was observed in the AOX-overexpressed strains. Compared with the wild type (WT), the polysaccharide production of AOXi25 was significantly increased by approximately 38%, while OE-AOX21 was significantly decreased by 80%. Six extracellular polysaccharides (EPSs) were extracted and purified from the WT, OE-AOX21, and AOXi25 strains. These EPSs, consisting of both neutral and acidic polysaccharides, were composed of five different monosaccharides in varying proportions. The average relative molecular masses were 1.68 × 10^3^, 2.66 × 10^3^, 1.67 × 10^3^, 2.42 × 10^3^, 1.12 × 10^3^, and 2.35 × 10^3^ kDa, respectively. Antioxidant assays demonstrated that all EPSs exhibited strong free radical scavenging activity with the acidic polysaccharide from AOXi25 showing the highest efficiency in ABTS^+^ scavenging. These findings highlight the significant role of AOX-derived ROS in regulating polysaccharide synthesis and accumulation in *Ganoderma lucidum*.

## 1. Introduction

*Ganoderma lucidum*, a member of the Basidiomycota class, is regarded as a valuable medicinal fungus. Its extract has been utilized for centuries in traditional Chinese medicine to promote overall health, improve immune function, and enhance vitality. The ongoing research into its bioactive compounds continues to highlight its potential therapeutic applications in modern medicine [1]. Numerous studies have demonstrated that *Ganoderma lucidum* contains a variety of bioactive compounds, including polysaccharides, triterpenoids, proteins, and steroids. These compounds have been shown to exhibit multiple therapeutic effects, such as inhibiting tumor cell growth, modulating immune function, and lowering blood glucose levels. The diverse bioactive constituents of *G. lucidum* contribute to its potential as an effective natural agent in the prevention and treatment of various diseases [2,3,4,5]. Among these polysaccharides are the primary bioactive components produced during *G. lucidum* fermentation. As a natural bioactive macromolecule composed of monosaccharides, polysaccharides possess strong immunomodulatory and antioxidant properties [6,7]. The biosynthesis of polysaccharides, as secondary metabolites, is influenced by various environmental factors, such as calcium ions, salicylic acid, and light [8,9]. Additionally, the structural characteristics of polysaccharides are closely linked to their biological activities [10,11]. However, the low polysaccharide content in *G. lucidum* limits its broader applications. This study explores the synthesis mechanism, structural characteristics, and antioxidant activity of these polysaccharides.

Alternative oxidase (AOX) is an enzyme that mediates cyanide-insensitive respiration. This mitochondrial pathway for cyanide-resistant respiration, present in plants, fungi, and protozoa, is encoded by the *AOX* nuclear gene. AOX facilitates the transfer of electrons from ubiquinone to oxygen, catalyzing its reduction to water [12,13]. The AOX pathway is characterized by the uncoupling of proton transport and ATP synthesis [14]. In addition, Alternative oxidase (AOX) plays a crucial role in mitigating the production of reactive oxygen species and has been studied extensively in plants, fungi, animals, and some prokaryotes. In deepwater rice, AOX is critical for anaerobic germination and growth [15]. In *Haematococcus pluvialis*, inhibiting alternative oxidase respiration promotes the carbon flow toward fatty acid and astaxanthin biosynthesis [16]. In *Arabidopsis thaliana*, AOX1a regulates nitrogen and carbon metabolism under low-nitrogen stress [17]. While AOX has been widely studied in these organisms, research on microorganisms, particularly filamentous fungi like *G. lucidum*, remains limited. As G. lucidum becomes an emerging model organism for basidiomycetes, understanding AOX’s role in metabolism is increasingly significant.

In addition, AOX is instrumental in controlling the generation of reactive oxygen species (ROS) in mitochondria [18]. By inhibiting electron transport chain complexes, AOX prevents excessive ROS generation, thereby protecting cells from oxidative damage [19]. Therefore, AOX is essential for reducing ROS levels. Research indicates that decreasing *AOX* gene expression in *G. lucidum* increases intracellular ROS levels, subsequently promoting ganoderic acid accumulation [20]. ROS can act as signaling molecules in fungi, inducing the expression of multiple genes to counteract oxidative stress. For example, adding farnesol during *Grifola frondosa* liquid fermentation elevates ROS levels, significantly boosting polysaccharide synthesis and yield. High ROS concentrations not only induce oxidative stress but also function as secondary messengers influencing secondary metabolite biosynthesis [21]. Understanding ROS mechanisms provides valuable insights into their role in fungal physiology and secondary metabolite regulation in medicinal fungi.

To investigate the relationship between *AOX* and *G. lucidum* polysaccharide synthesis, the *AOX* gene was cloned. RNA interference (RNAi) was used to construct silenced strains, while high-expression strains were generated with a promoter-driven *AOX* gene. This study investigated the impact of the *AOX* gene on the synthesis of *Ganoderma lucidum* polysaccharides, analyzing the underlying mechanisms with a focus on intracellular ROS levels and the key enzymes involved in polysaccharide biosynthesis. Additionally, the effects of the *AOX* gene on the physicochemical properties, structure, and biological activity of polysaccharides synthesized by engineered strains were examined.

## 2. Materials and Methods

### 2.1. Strains and Culture Conditions

*G. lucidum* strain (Jinzhi No. 8), provided by the Institute of Quartermaster Engineering and Technology, was used as the WT strain. The wild-type, AOX-overexpressed strains and AOX-silenced strains were cultured in yeast extract medium (YM) containing 2% (*w*/*v*) glucose, 0.1% KH_2_PO_4_, 0.1% MgSO_4_·7H_2_O, and 0.5% yeast extract at 28 °C. E. coli (DH5a) was cultured in Luria Broth (LB) medium supplemented with ampicillin (100 μg/mL) at 37 °C for plasmid amplification.

### 2.2. Construction of AOX-Silenced and AOX-Overexpressed Strains

The vector was constructed by homologous recombination method. pAN7-1 vector was used as the backbone, with BamHI and HindIII sites selected for double digestion. Genomic DNA was extracted from *Ganoderma lucidum* using the CTAB method, and total RNA was isolated using the OMEGA kit, followed by reverse transcription to obtain cDNA. Primers were designed to amplify hygromycin gene for screening successful transformants. The primers, which are used for amplifying gene fragments, are shown in Appendix A. The fungal RNA interference (RNAi) vector, pAN7-dual-AOXi, was constructed to inhibit AOX expression. This vector is driven by the 35S and glyceraldehyde-3-phosphate dehydrogenase (gpd) promoters. Additionally, the pAN7-OE-AOX vector, driven by the gpd promoter, was used for *AOX* gene overexpression. The silenced and overexpressed vectors were introduced into *Ganoderma lucidum* protoplasts via polyethylene glycol (PEG)-mediated transformation. The overexpressed and silenced strains were designated as OE-AOX and AOXi, respectively.

### 2.3. Analysis of Gene Expression by Real-Time PCR

The mycelia of *G. lucidum* was cultured on YM solid plate at 28 °C for 7 days, followed by fermentation in liquid YM medium for 3 days. Samples were then collected for the analysis of ROS-related genes and polysaccharide synthesis genes. Fungal RNA Kit (OMEGA, China) was used to extract total RNA. Gene expression levels of the target genes in each sample were quantified by calculating the difference between the threshold cycle (CT) value of the analyzed gene and the CT value of the GAPDH housekeeping gene. The number of gene expression changes was calculated using the 2^−ΔΔCT^ method [22]. The primers used for Real-Time PCR amplification of gene fragments are listed in Appendix A.

### 2.4. Chemical Analysis

The active oxygen content was analyzed using the fluorescence probe 2′,7′ -dichlorofluorescein diacetate (DCFH-DA) staining method. DCFH-DA solution was added to both the *G. lucidum* mycelia and mycelia suspension, followed by incubation at 37 °C for 30 min. The *G. lucidum* mycelia were then sliced and imaged under a laser scanning confocal microscope. The mycelia suspension was measured using a microplate reader, and the intensity of DCF fluorescence was compared. Polysaccharide content was determined using the phenol-sulfuric acid method [23]. The mycelium was removed from the fermentation broth by centrifuged at 12,000 rpm for 10 min. The supernatant is mixed with 95% ethanol at a ratio of 1:4, maintained at 4 °C for 24 h, fully washed with 95% ethanol, and freeze-dried to obtain the extracellular crude polysaccharide of the fermentation broth. BCA protein kit was used to quantified the protein content.

### 2.5. Polysaccharide Purification

Proteins were removed from crude polysaccharides using the Sevag method and then purified by DEAE-650M cellulose anion exchange chromatography. The collected polysaccharide was dialyzed using a rotary evaporator with a molecular weight cut-off dialysis bag (Solarbio MD55 7000D, China, Beijing), followed by vacuum freeze-drying. Pure polysaccharides with different components were obtained based on the variation in ionic groups and their quantities.

### 2.6. Structural Characterization of Polysaccharides

#### 2.6.1. UV–Vis and FT-IR Spectra Analysis

The UV–Vis absorption spectra of each polysaccharide sample were measured using a spectrophotometer (Eppendorf Biospectrometer, Germany, Hamburg) within the wavelength range of 200–400 nm [24]. Then, 1 mg of each polysaccharide sample was ground with 150 mg of KBr, pressed into a transparent pellet, and analyzed using an FT-IR system (Thermo Fisher Nicolet IS50, Waltham, MA, USA) to determine the functional groups present. The analysis was performed with 32 scans and a resolution of 0.5 cm^−1^ [25].

#### 2.6.2. Scanning Electron Microscope (SEM) Assay

The microstructures of each polysaccharide sample were analyzed using a scanning electron microscope (FEI Company, Hillsboror, OR, USA). Prior to imaging, the samples were gold-coated under vacuum using ion sputtering and subsequently observed and photographed under the microscope. The analysis was conducted at a voltage of 10 kV and a magnification of 1000×.

#### 2.6.3. Determination of Monosaccharide Composition and Molecular Weight

The homogeneity and molecular weight (MW) of polysaccharides were determined by High-Performance Size Exclusion Chromatography (HPSEC) with a TSK-Gel G4000 PWXL column (30 cm × 7.5 mm) and a refractive index detector. The mobile phase consisted of ultra-pure water, with a flow rate of 0.5 mL/min. The injection volume was 20 μL, and the column and detector were maintained at 50 °C. Dextran standards of various molecular weights were used to construct a standard curve. The molecular weight of polysaccharides was estimated based on the retention time and the lgMw from the dextran standards curve [26].

### 2.7. In Vitro Antioxidant Effects

#### 2.7.1. ABTS^+^ Radical Scavenging Activity

The ABTS^+^ radical scavenging activity of polysaccharides was assessed using a modified version of a previously reported method [7]. Briefly, equal volumes of 2.6 mmol/L K_2_S_2_O_8_ solution and 7.4 mmol/L ABTS^+^ solutions were mixed and kept in the dark at room temperature 25 °C for 16 h. The ABTS^+^ reaction mixture was then adjusted to an absorbance of 0.70 ± 0.02 at 734 nm. A 30 μL aliquot of each polysaccharide sample, with concentrations ranging from 0.5 to 2.5 mg/mL, was added to 180 μL of ABTS^+^ solution and mixed vigorously. Ascorbic acid (VC) at the same concentration was used as a positive control. After reacting at room temperature 25 °C for 6 min, the absorbance was measured at 734 nm using a microplate reader. The ABTS^+^ radical scavenging activity was calculated using the following Formula (1):(1)$${\mathrm{ABTS}}^{+}\;\mathrm{radical}\;\mathrm{scavenging}\;\mathrm{activity}\;\left(\;\%\;\right)=\left[1-\frac{A_{1}-A_{2}}{A_{0}}\right]\times 100\%$$
where *A*_0_ is the absorbance of the control without the sample, *A*_1_ is the absorbance of the sample, and *A*_2_ is the absorbance of the sample blank without ABTS^+^.

#### 2.7.2. Hydroxyl Radical Scavenging Activity

The hydroxyl radical scavenging activity was determined using a modified Fenton reaction method [27]. The experiment consisted of three groups: the H_2_O_2_ oxidative injury group, the non-injury group, and the sample group. Briefly, in the sample group, 1 mL of FeSO_4_ (2 mmol/L) and 1 mL of 1,10-phenanthroline (2 mmol/L) were added to 400 μL of the polysaccharide sample (0.5–2.5 mg/mL). Then, 600 μL of PBS buffer and 600 μL of 0.1% H_2_O_2_ were added to initiate the reaction, which was incubated at 37 °C for 30 min. The absorbance was measured at 510 nm, with VC used as a positive control. The hydroxyl radical scavenging activity was calculated using the following Formula (2):(2)$$\mathrm{OH}\;\mathrm{radical}\;\mathrm{scavenging}\;\mathrm{activity}\;\left(\%\right)=\frac{A_{1}-A_{2}}{A_{3}-A_{2}}\times 100\%$$
where *A*_1_ is the absorbance of the sample group, *A*_2_ is the absorbance of the injury group (water instead of the sample), and *A*_3_ is the absorbance of the non-injury group (water instead of H_2_O_2_).

#### 2.7.3. DPPH Radical Scavenging Activity

The DPPH radical scavenging activity was determined using a modifying version of the method described by Shen et al. [7]. Briefly, 1 mL DPPH solution (0.2 mmol/L DPPH in DMSO) was mixed with 1 mL of polysaccharide sample solutions at different concentrations (0.5–2.5 mg/mL). Ascorbic acid (VC) at the same concentration was used as a positive control. The reaction mixture was incubated in the dark for 30 min, and the absorbance at 525 nm was measured using a microplate reader (USA, California, San Jose). The DPPH radical scavenging activity and the scavenging rate of different samples were calculated using the following Formula (3):(3)$$\mathrm{DPPH}\;\mathrm{radical}\;\mathrm{scavenging}\;\mathrm{activity}\;\left(\%\right)=\left[1-\frac{A_{1}-A_{2}}{A_{0}}\right]\times 100\%$$
where *A*_1_ is the absorbance of the polysaccharide and DPPH solution, *A*_2_ is the absorbance of the polysaccharide and DMSO solution, and *A*_0_ is the absorbance of DPPH solution.

### 2.8. Statistical Analysis

All the results presented in this dissertation were obtained from three independent samples. The mean of each sample was analyzed using an independent-samples t-test and analysis of variance (ANOVA), performed with SPSS statistical software (version 22.0). The significance level of *p* < 0.05 was applied.

## 3. Results and Discussion

### 3.1. Construction of AOX-Silenced Strains and AOX-Overxepressed Strains

To explore the role of ROS in *G. lucidum*, AOX-silenced strain was constructed by a double promoter subsystem, and the AOX-overexpressed strains was constructed by a strong promoter. Transformants were screened by hygromycin resistance selectable marker. The construction of AOX-silenced and overexpressed vectors is illustrated in Appendix A.

Positive transformants were selected on plates containing hygromycin as the selectable marker. Using *GAPDH* as the reference gene, the transcription levels of the *AOX* gene was quantified through real-time PCR. AOX-silenced strains (AOXi3 and AOXi25) and AOX-overexpressed strains (OE-AOX2 and OE-AOX21) were successfully identified. In comparison with the WT, AOX gene expression in silenced strains AOXi3 and AOXi25 was reduced by approximately 86% and 68%, respectively, whereas in overexpressed strains OE-AOX2 and OE-AOX21, expression increased by about 105% and 68%, respectively (Figure 1).

### 3.2. Analysis of Intracellular ROS Level

Prior studies have shown that *AOX* is involved in maintaining the balance of electron transport in mitochondria, and ROS in cells are closely linked to electron transport. To study the effect of *AOX* on ROS level in *G. lucidum*, the fluorescence intensity of the hyphae from engineering strain was observed by laser confocal microscopy. The fluorescence intensities of hyphae from the two AOX-silenced strains were significantly higher than those of the wild type, indicating that silencing the *AOX* gene induces ROS production and accumulation in *G. lucidum* cells. However, the fluorescence intensities of hyphae from the AOX-overexpressed strains were reduced to varying degrees (Figure 2). To further quantify ROS levels, we analyzed AOXi3, AOXi25, OE-AOX-2, and OE-AOX-21 strains. Fluorescence probe staining revealed that *AOX* gene silencing resulted in a 1.78 and 1.96 fold increase in fluorescence intensity relative to the wild type, while fluorescence intensity in AOX-overexpressed strains was 0.56 and 0.47 fold compared to the WT strain (Figure 3). Increased ROS levels are known to induce intracellular oxidative stress. We assessed the expression of genes associated with the intracellular antioxidant system, including superoxide dismutases (SOD1 and SOD4, from the Cu/Zn-SOD family, and SOD2 from the Mn-SOD family), catalases (CAT1 and CAT2), glutathione peroxidase (GPX), and ascorbate peroxidase (APX). The expression levels of CAT1, SOD1, SOD2, SOD4, APX, and GPX were upregulated to varying degrees in AOX-silenced strains, while their expression levels were downregulated to different degrees in the overexpression strains, except for SOD1 (Figure 4). These findings demonstrate that interference with the *AOX* gene activates the antioxidant system in *G. lucidum* cells and alters the activity levels of antioxidant enzymes. This suggests that these enzymes act synergistic, potentially due to the effects of ROS on different enzymatic targets. Collectively, these results indicate that *AOX* has a significant role in the regulation of intracellular ROS levels in *G. lucidum*.

### 3.3. Influences of AOX on EPS Synthesis

Figure 5A shows the production of extracellular polysaccharides (EPSs) by liquid fermentation of *G. lucidum*. During shake-flask fermentation, the EPS production of AOXi strains was significantly higher than the wild type. After 96 h fermentation, the EPS yield of AOXi25 stain reached 3.74 ± 0.12 g/L, which is 1.38 times higher than WT. In contrast, the EPS yields of the overexpressed strains (OE-AOX2 and OE-AOX21) were 0.58 ± 0.09 g/L and 0.53 ± 0.04 g/L, respectively. To further study the synthesis of extracellular polysaccharides, two key polysaccharide synthesis enzyme genes, *PMI* and *PGM*, were selected for analysis in WT, OE-AOX2, OE-AOX21, AOXi3, and AOXi25 strains. The expression levels of *PMI* and *PGM* were higher in AOX-silenced strains compared to WT, whereas their expression was downregulated in the overexpressed strains (Figure 5B). These suggest that *AOX* influences the synthesis of *G. lucidum* polysaccharide. As shown in Figure 5C, there was a significant positive correlation between EPS yield and ROS levels (R^2^ = 0.77), indicating that *AOX* may directly affect *G. lucidum* polysaccharide synthesis by inducing ROS production. These results indicate that the *AOX* gene plays a vital role in maintaining the balance of ROS in *Ganoderma lucidum*. Silencing *AOX* expression increases the intracellular ROS level, while its overexpression disrupts ROS synthesis and affects polysaccharide production.

### 3.4. Purification and Structural Characteristics of Polysaccharides

#### 3.4.1. Purification of Polysaccharides

The OE-AOX2 and AOXi25 strains demonstrated high efficiency among the overexpressed and silenced strains examined in the previous study. Therefore, these two strains were selected for further investigation. As shown in Figure 6, WT, OE-AOX2, and AOXi25 strains all yielded two polysaccharide components through gradient elution: acidic polysaccharide and neutral polysaccharide. The separation mechanism of anion exchange chromatography involves not only ion exchange but also adsorption and desorption processes. Different polysaccharide components exhibit varying degrees of adsorption to the exchanger. Acidic polysaccharides are adsorbed onto the exchanger, while neutral polysaccharides are not and are eluted first. This allows the separation of acidic and neutral polysaccharide components. The acid polysaccharides from WT, OE-AOX2, and AOXi25 were named WE-A, OE-A, and iE-A, respectively, while the neutral polysaccharides were designated WE-N, OE-N, and iE-N. Subsequently, six polysaccharide fractions were obtained through concentration, dialysis, and lyophilization for further analysis.

#### 3.4.2. UV Analysis and FT-IR Spectra Analysis

The UV spectra of WE-N, WE-A, OE-N, OE-A, iE-N, and iE-A showed no absorption peaks at 280 nm and 230 nm (Figure 7A), indicating that the sample contained almost no protein. This result is consistent with the low protein content of the crude polysaccharide.

Infrared spectroscopy (FT-IR), an essential method for analyzing the molecular structure of organic compounds, is widely used for structural analysis of sugar compounds. The FT-IR spectra of WE-N, WE-A, OE-N, OE-A, and iE-N revealed similar polysaccharides absorption peaks within the ranges of 4000 cm^−1^ to 400 cm^−1^ (Figure 7B). The peak near 3400 cm^−1^ corresponds to the characteristic stretching vibration of hydroxyl groups (-OH) in sugar molecules [28], while the signal at 2926 cm^−1^ represents the stretching vibration of the C-H bond. Additionally, a stretching peak at 1651 cm^−1^ and a weak stretching peak at 1412 cm^−1^ are characteristic signals of deprotonated carboxyl groups, indicating the presence of uronic acid in the polysaccharides structure [29]. No obvious peak at 1616 cm^−1^ indicates that proteoglycans are not present in the purified polysaccharide fractions. The absorption peaks at 1419 cm^−1^ and 1379 cm^−1^ were caused by carbonyl stretching vibration in carboxylate ions, further confirming the presence of carboxyl groups in all six polysaccharide components. The absorption peak at 870 cm^−1^, caused by the variable-angle vibration of CH_2_ bonds, indicates that all six polysaccharides contain β-D-glucopyranose rings, with their molecules primarily linked by β-glycosidic bonds. In conclusion, the six polysaccharide fractions are β-glycosidic-linked polysaccharides that contain no protein and no amide groups.

#### 3.4.3. Analysis of Morphological Property

Scanning electron microscopy (SEM) enables high-resolution micro-area morphological analysis and is widely applied in various fields such as materials science, medicine, and food. As shown in Figure 8, after gold-spraying treatment, the surface of WE-N, WE-A, OE-N, OE-A, iE-N, and iE-A polysaccharides mainly present a loose fragment-like aggregation state that forms sheet-like, layered structures closely stacked together. The images clearly illustrate the amorphous structure of *G. lucidum* polysaccharides. The acidic polysaccharides WE-A, OE-A, and iE-A display loose and porous surfaces with small spherical and filamentous structures, which may be associated with their physical and chemical properties.

#### 3.4.4. Analysis of Basic Physicochemical Property

The molecular weight, type, and position of the functional groups such as hydroxyl, sulfate, amino, carboxyl, and phosphate groups, the type of saccharide and glycosidic branching, and the degree of substitution of polysaccharides are closely related to their biological activity [30]. Using the calibration curve established with standard dextran, the molecular weights of WE-N, WE-A, OE-N, OE-A, iE-N, and iE-A were estimated to be 1.68 × 10^3^, 2.66 × 10^3^, 1.67 × 10^3^, 2.42 × 10^3^, 1.12 × 10^3^, and 2.35 × 10^3^ kDa, respectively. Overall, acidic polysaccharides have higher molecular weights than neutral ones, and wild-type polysaccharides have higher molecular weights compared to those from engineering strains. The result indicates that *AOX* gene influences polysaccharides molecular weight during the regulation of polysaccharide synthesis. The monosaccharide composition of polysaccharides is an important parameter for understanding the relationship between physicochemical properties and structure [31]. As shown in Table 1, WE-N, WE-A, OE-N, OE-A, iE-N, and iE-A were made up of five monosaccharides, including ribose, arabinose, mannose, galactose, and glucose. The percentage of arabinose and ribose were relatively low across all samples. The polysaccharides were primarily made up of mannose, galactose, and glucose. In addition, glucose was the most abundant monosaccharide in neutral polysaccharides, while mannose dominated in acidic polysaccharides. The formation of acidic polysaccharides may result from chemical modifications, such as the carboxylation, sulfation, or phosphorylation of monosaccharides during polysaccharide synthesis. These modifications introduce negatively charged groups, thereby rendering monosaccharides acidic [32]. Furthermore, studies have shown that polysaccharides with high mannose content exhibit greater antioxidant activity than those with high glucose content [33].

### 3.5. In Vitro Antioxidant Activity

#### 3.5.1. ABTS^+^ Radical Scavenging Ability

In recent years, polysaccharides have attracted considerable interest because of their powerful antioxidant activity and low incidence of side effects, positioning them as valuable therapeutic agents [34]. The ABTS+ radical assay is commonly employed to assess the total antioxidant capacity of compounds. The ABTS+ radical scavenging activities of WE-N, WE-A, OE-N, OE-A, iE-N, and iE-A, compared to VC, are presented in Figure 9A. The free radical scavenging capacity was observed to increase with higher concentrations of polysaccharides. The ABTS^+^ radical scavenging ability of WE-N, WE-A, OE-N, OE-A, iE-N, and iE-A were significantly lower than that of the VC group at concentrations below 0.5 mg/mL. However, at the concentration of 2.5 mg/mL, the acidic polysaccharides demonstrated a stronger scavenging effect on ABTS^+^ radical compared to neutral polysaccharide, which might be attributed to the presence of more uronic acid and sulfate [35,36]. Furthermore, the ABTS^+^ radical scavenging rates of WE-A, OE-A, and iE-A reached 94.26 ± 4.33%, 85.07 ± 0.84%, and 88.62 ± 2.61%, respectively. The strong ABTS^+^ radical scavenging ability of *G. lucidum* polysaccharides may be attributed to their capacity to convert reactive free radicals into stable forms and terminate chain reactions by donating electrons [37]. Previous studies have reported that acidic polysaccharides exhibit higher biologically active than neutral polysaccharides [38]. Consistent with these findings, the results showed that the scavenging activities of the acidic polysaccharides WE-A, OE-A, and iE-A were higher than those of the neutral polysaccharides against ABTS^+^ radicals.

#### 3.5.2. Hydroxyl Radical Scavenging Ability

The hydroxyl radical is the most reactive and toxic free radical, capable of reacting with nearly all molecules in living cells at extremely rapid rate. These molecules include amino acids, phospholipids, nucleic acids, and organic acids. Therefore, scavenging hydroxyl radicals plays a crucial role in the antioxidant defense mechanisms of cellular systems. As shown in Figure 9B, as the concentration of the six polysaccharides increased, the hydroxyl free radical scavenging rate also increased. When the concentration reached 2.5, the scavenging rate stabilized and leveled off. Uronic acid content plays a key role in chelating Fe^2+^, thereby reducing hydroxyl radical generation and enhancing the reducing ability of polysaccharides. This enhances the hydroxyl radical scavenging activity of *Ganoderma lucidum* polysaccharides. Polysaccharides that contain higher levels of uronic acid show increased hydroxyl radical scavenging activity. The hydroxyl radical scavenging rates of acidic and neutral polysaccharides from AOX-silenced and AOX-overexpressed strains were higher than those from the WT. Specifically, the maximum scavenging rates of acidic polysaccharides iE-A and OE-A reached 52.08 ± 0.72% and 46.69 ± 6.61%, respectively, compared to 34.46 ± 1.01% in the WT. These findings suggest that AOX-silenced strains may contain higher levels of uronic acid than AOX-overexpressed.

#### 3.5.3. DPPH Radical Scavenging Ability

The DPPH radical is a stable free radical at room temperature and is widely utilized to evaluate the free radical scavenging potential of antioxidants. In this method, DPPH radicals are scavenged by protons provided by antioxidants, forming a non-radical product, DPPH-H, which terminates the radical chain reaction [39]. The DPPH radical scavenging abilities of WE-N, WE-A, OE-N, OE-A, iE-N, iE-A, and VC are shown in Figure 9C. Polysaccharides can form stable DPPH complexes, but their scavenging ability is lower than that of VC. As shown in Figure 9C, the DPPH radical scavenging activity of the six polysaccharides gradually increased at concentrations ranging from 0 to 0.5 mg/mL. The scavenging ability reached a plateau with minimal fluctuations between concentrations of 0.5 and 2.5 mg/mL. Among the six polysaccharides, iE-A exhibited a relatively high DPPH scavenging ability, reaching its peak at a concentration of 1.0 g/mL. At 2.5 mg/mL, the DPPH radical scavenging rates were as follows: WE-A (62.43 ± 7.67%), OE-A (59.64 ± 6.93%), iE-A (58.58 ± 0.88%), iE-N (50.11 ± 1.98%), WE-N (49.42 ± 1.19%), and OE-N (41.60 ± 0.76%). These results indicate that acidic polysaccharides have a stronger DPPH radical scavenging ability compared to neutral polysaccharides, likely due to their superior hydrogen-donating capacity. In summary, *Ganoderma lucidum* polysaccharides may donate hydrogen atoms to react with DPPH radicals, thereby converting them into more stable products and effectively scavenging DPPH radicals.

## 4. Conclusions

Our results showed that the mRNA levels of enzymes involved in the intracellular antioxidant system were significantly increased in AOX-silenced strains, which corresponded with elevated intracellular ROS levels. In contrast, the results were the opposite in AOX-overexpressed strains. This suggests that *AOX* plays a role in ROS accumulation and helps maintain the dynamic balance of intracellular ROS. In this study, six polysaccharides (WE-N, WE-A, OE-N, OE-A, iE-N, and iE-A) were extracted and purified from WT, AOX-overexpressed and AOX-silenced strains. The expression of *AOX* significantly affected the surface morphology, chemical composition, monosaccharide composition ratio, and molecular weight of the polysaccharides. Antioxidant assays revealed that all six polysaccharides exhibited scavenging activity against ABTS^+^ and moderate scavenging activity against DPPH radicals and hydroxyl radicals. Additionally, the scavenging activities of acidic polysaccharides were consistently better than those of the neutral polysaccharides. The above results suggest that ROS, influenced by the *AOX* gene, mediates the secondary regulation of *G. lucidum* polysaccharides and alters their antioxidant activity. Collectively, our study provides insight into a novel metabolic regulation mechanism targeting ROS.

## Figures and Tables

**Figure 1 foods-14-00826-f001:**
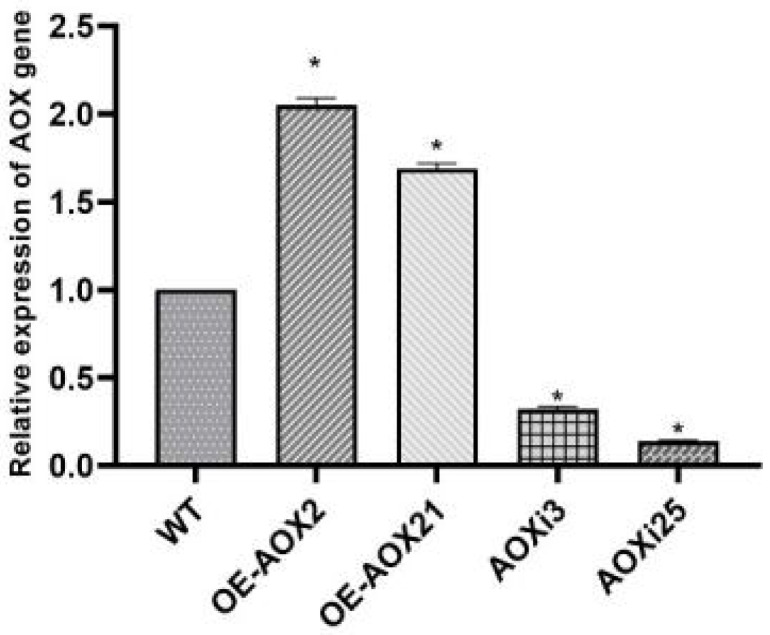
Expression levels of *AOX* in WT, AOX-overexpressed and AOX-silenced strains. The asterisk (*) indicates statistical significance of the results.

**Figure 2 foods-14-00826-f002:**
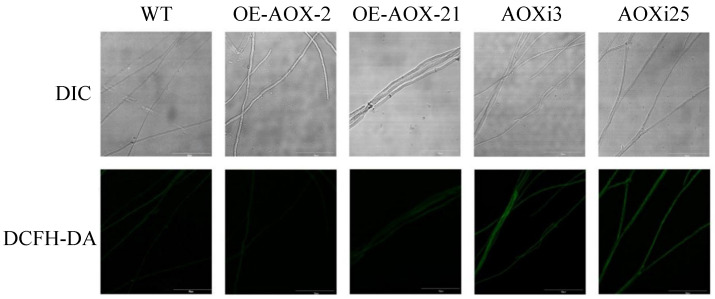
Fluorescence microscopy of the mycelia from WT, AOX-overexpressed and AOX-silenced strains.

**Figure 3 foods-14-00826-f003:**
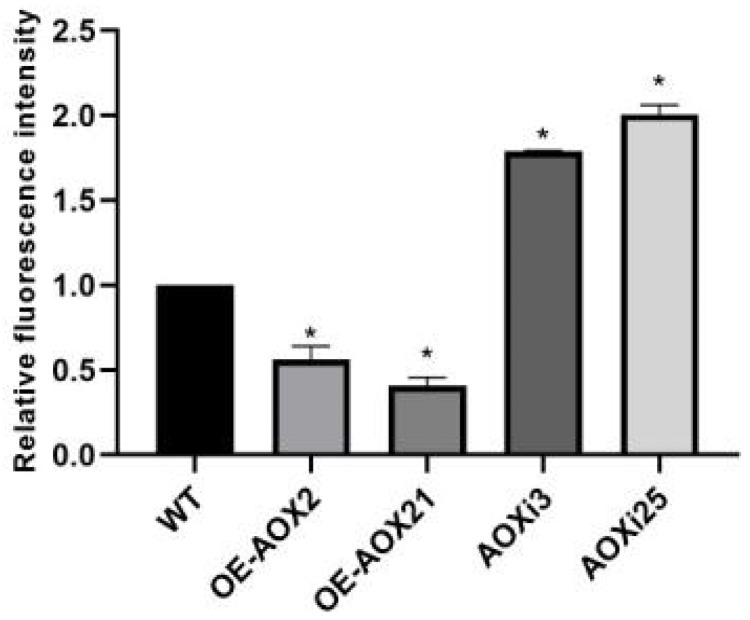
Quantitative analysis of fluorescence intensity of WT, AOX-overexpressed and AOX-silenced strains. The asterisk (*) indicates statistical significance of the results.

**Figure 4 foods-14-00826-f004:**
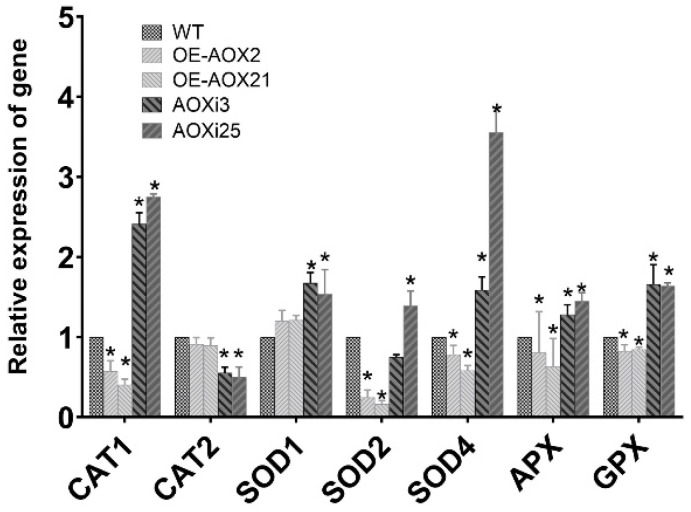
Real-time PCR analysis of antioxidant-system-related gene expression in WT, AOX-overexpressed and AOX-silenced strains. The asterisk (*) indicates statistical significance of the results.

**Figure 5 foods-14-00826-f005:**
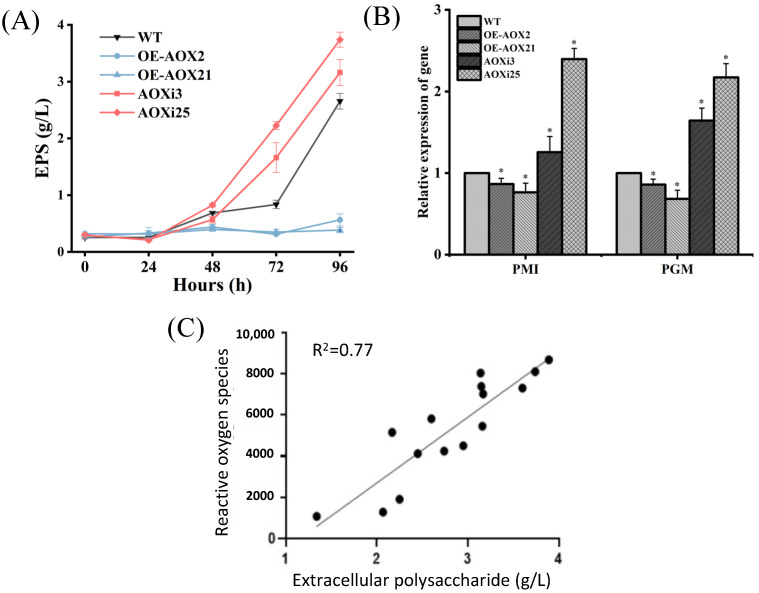
(**A**) Changes of polysaccharide yield in WT, AOX-overexpressed and AOX-silenced strains; (**B**) expression levels of polysaccharide synthesis genes PMI and PGM in WT, AOX-overexpressed and AOX-silenced strains; (**C**) correlation between ROS levels and EPS production. The asterisk (*) indicates statistical significance of the results.

**Figure 6 foods-14-00826-f006:**
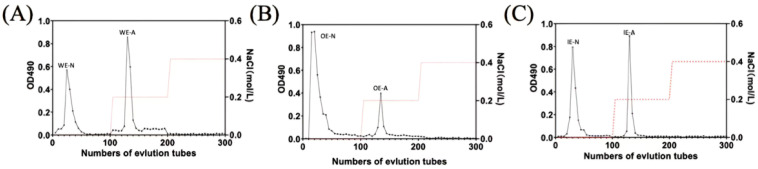
Stepwise elution curves of polysaccharides from WT (**A**), OE-AOX2 (**B**), and AOXi25 (**C**) using a DEAE-650M chromatography column.

**Figure 7 foods-14-00826-f007:**
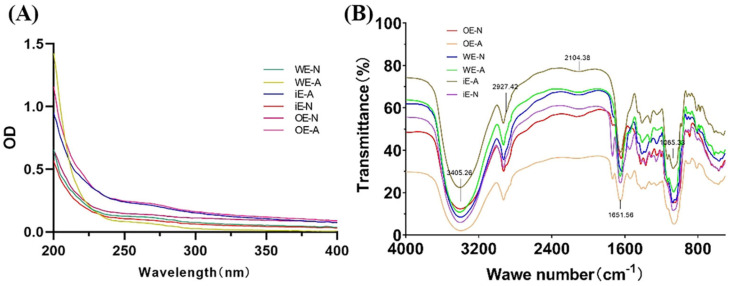
The UV (**A**) and FT-IR (**B**) spectra of WE-N, WE-A, OE-N, OE-A, iE-N, and iE-A.

**Figure 8 foods-14-00826-f008:**
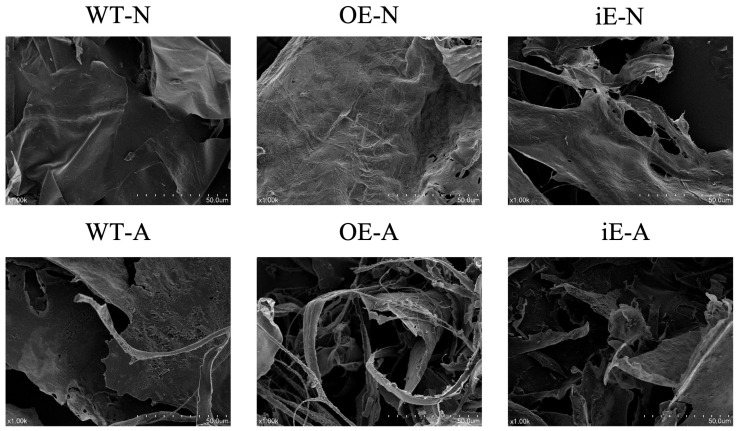
The SEM images of WT-N, WT-A, OE-N, OE-A, iE-N, and iE-A.

**Figure 9 foods-14-00826-f009:**
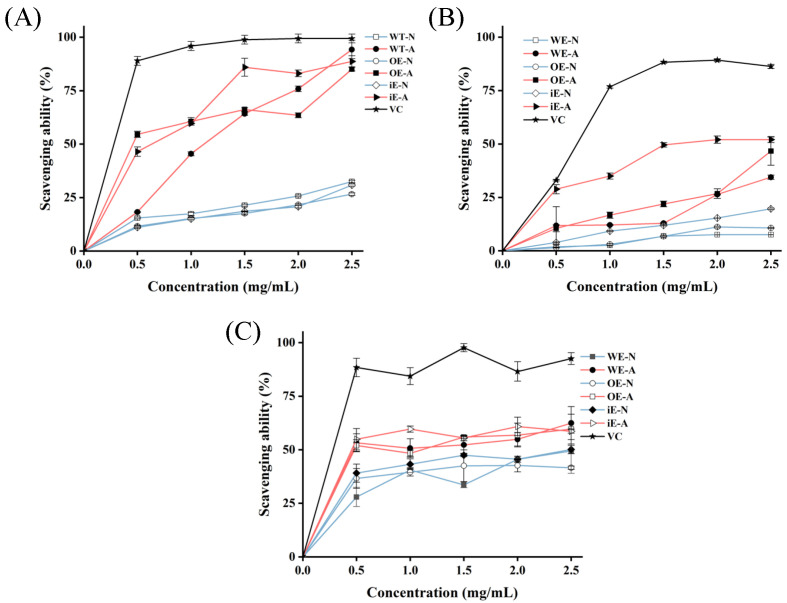
Antioxidant activities of WE-N, WE-A, OE-N, OE-A, iE-N, and iE-A ABTS^+^ radical scavenging activity (**A**), hydroxyl radical scavenging activity (**B**), and DPPH radical scavenging activity (**C**).

**Table 1 foods-14-00826-t001:** The basic physicochemical properties of WE-N, WE-A, OE-N, OE-A, iE-N, and iE-A.

Basic Physicochemical Properties		Sample Name					
		WE-N	WE-A	OE-N	OE-A	iE-N	iE-A
Average molecular weights (kDa)		1.68 × 10^3^	2.66 × 10^3^	1.67 × 10^3^	2.42 × 10^3^	1.12 × 10^3^	2.35 × 10^3^
Monosaccharide composition (%)	Ribose	2.54 ± 0.01	4.23 ± 0.11	2.82 ± 0.23	5.35 ± 0.15	3.55 ± 0.09	6.22 ± 0.08
	Arabinose	3.19 ± 0.29	1.19 ± 0.05	3.82 ± 0.28	2.26 ± 0.20	16.35 ± 0.38	0.76 ± 0.02
	Mannose	30.03 ± 0.51	70.05 ± 0.91	16.57 ± 0.71	54.94 ± 0.95	7.64 ± 0.32	48.27 ± 0.46
	Galactose	10.17 ± 0.05	4.25 ± 0.07	27.70 ± 0.01	17.11 ± 0.35	14.38 ± 0.61	6.16 ± 0.12
	Glucose	53.52 ± 0.32	20.29 ± 0.24	49.09 ± 0.62	31.98 ± 0.95	73.19 ± 0.98	38.59 ± 0.22

## Data Availability

The original contributions presented in this study are included in the article.

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
