# Peer review of "AOX Affects the Synthesis of Polysaccharides by Regulating the Reactive Oxygen Species in Ganoderma lucidum"

_foods, 2025, doi:10.3390/foods14050826_

Round 1
Reviewer 1 Report
Comments and Suggestions for Authors
Line 22. The number 3 is superscript in 1.68x103.
In the figure caption of the supplementary material, place the word "(gpd or gpdA)" after the text "glyceraldehyde-3-phosphate dehydrogenase".
Lines 98-99. To homogenize, gpd or gpdA in the figure caption and the manuscript.
Lines 122-125. How were the crude polysaccharides from mycelium extracted? It is not clear. Are they soluble in water?
Lines 129-130. What is the cut-off dialysis bag? It is important.
Lines 137-138. What is the number of scans and resolution in the FTIR analysis?
Lines 140-143. What is the voltage and the magnification used in the SEM? .
Line 155. Typing mistake in the sub-section (2.7.1. .)
Line 175. VC or Vc.
Line 180. Typing mistake in the sub-section (2.7.1. .)
Line 210. Figure 1B should be together 1D. Graphs with the same units in the y-axis.
Line 220. Figure 1A should be alone. Only images.
Line 224. Figure 1C should be alone.
Line 230. Typing mistake (strains. while).
Lines 261-264. The three graphs present different information but are together.
Line 271. It is not clear what the criteria for the selection of strains are. Relative expression of genes? Protein content?, Polysaccharide content?
Lines 290-293. The sentence "The UV spectra of WE-N, WE-A, OE-N, OE-A, iE-N and iE-A showed no absorption peaks at 280 nm and 230 nm (Figure 4A), indicating that the sample contained almost no protein. This result is consistent with the protein content determination of the crude polysaccharide" is contrary to the previous sentence (lines 269-270) "The protein contents of crude polysaccharides from the WT, OE-AOX2 and AOXi25 strains were 1.62%, 3.21% and 10.73%, respectively, as determined using the BCA protein assay kit."
Lines 310-311. The sentence "A deeper examination of the infrared spectra reveals that polysaccharides share many similar characteristic absorption peaks within this spectral range." is incomplete because the discussion does not continue (not show graph or indicate the peaks).
Lines 323-328. Rewriting the sentence "Additionally, a relationship was observed between the polysaccharides structure and....... the physical structure of G. lucidum polysaccharides." to relate to the previous sentence or removing.
Lines 331-348. The sugars described in section 3.4.3 and Table 1 are neutrals. It would be important to discuss why they form acidic polysaccharides.
Line 355. Vitamin C was defined as Vc. Remove the words "vitamin C".
Lines 351-368. If the acidic polysaccharides have high ABTS+ radical scavenging abilities, it is essential to discuss why, according to the results, they are composed only of neutral sugar.
Lines 369-385. In the section 3.5.2. (Hydroxyl radical scavenging ability), Why suggest that only AOX-silenced strains may contain higher uronic acid levels if both AOX-silenced and AOX-overexpression strains have higher antioxidant activity than the wild type?
Generally, there is a lack of references and details regarding the material and methods.
Author Response
Thank you very much for taking the time to review this manuscript. Please find the detailed responses below and the corresponding revisions/corrections highlighted/in track changes in the re-submitted files.
Comments 1: Line 22. The number 3 is superscript in 1.68x103.
Response 1: The format problem has been revised in the manuscript.
Comments 2: In the figure caption of the supplementary material, place the word "(gpd or gpdA)" after the text "glyceraldehyde-3-phosphate dehydrogenase".
Response 2: gpd have been add in the supplementary material.
Comments 3: Lines 98-99. To homogenize, gpd or gpdA in the figure caption and the manuscript.
Response 3: The figure caption and the manuscript have been homogenized.
Comments 4: Lines 122-125. How were the crude polysaccharides from mycelium extracted? It is not clear. Are they soluble in water?
Response 4: This manuscript focuses on studying the polysaccharide group in the fermentation broth, without extracting the crude polysaccharides from the mycelium. Additionally, the crude polysaccharides are dissolved in water. The detailed methodology for extracting the crude polysaccharides from the fermentation broth has been revised and is now provided.
Comments 5: Lines 129-130. What is the cut-off dialysis bag? It is important.
Response 5: The exact name of the “dialysis bag” is dialysis membrane, and the dialysis membrane model used in this experiment is MD55 (7000D, link: https://www.solarbio.com/goodsInfo?id=11011), produced by Solarbio (Beijing, China).
It is very important. High salt concentrations from adversely affects the structure or biological activity of the polysaccharide. During polysaccharide purification, the salt concentration in the solution may be high, particularly during processes such as salting out or solvent exchange. Dialysis bags are used to progressively remove salts and low-molecular-weight compounds from the solution, thereby helping to regulate ion concentrations.
Comments 6: Lines 137-138. What is the number of scans and resolution in the FTIR analysis?
Response 6: The manuscript has been revised.
Comments 7: Lines 140-143. What is the voltage and the magnification used in the SEM? .
Response 7: The analysis was conducted at a voltage of 10 kV and a magnification of 1,000×. The manuscript has been revised.
Comments 8: Line 155. Typing mistake in the sub-section (2.7.1. .)
Response 8: The typing mistake have been corrected.
Comments 9: Line 175. VC or Vc.
Response 9: VC is the abbreviation of ascorbic acid. We have uniformed the use of “VC”.
Comments 10: Line 180. Typing mistake in the sub-section (2.7.1. .)
Response 10: The typing mistake have been corrected.
Comments 11: Line 210. Figure 1B should be together 1D. Graphs with the same units in the y-axis.
Response 11: After vector transformed and screened by hygromycin resistance selectable marker, there are many colonies on the plate. But the expression of AOX gene is different in Ganoderma lucidum. Figure 1B shows the strains screened with significant differences in AOX gene expression by qPCR, while Figure 1D presents the differential expression of genes related to the antioxidant system in selected AOX overexpressed (OE-AOX2 and OE-AOX21) and AOX-silenced (AOXi3 and AOXi25) strains. Although both figures share the same y-axis, they represent distinct aspects of the study and should be presented separately.
Comments 12: Line 220. Figure 1A should be alone. Only images.
Line 224. Figure 1C should be alone.
Response12: Figure 1A and Figure 1C have been alone.
Comments 13: Line 230. Typing mistake (strains. while).
Response13: The typing mistake have been corrected.
Comments 14: Lines 261-264. The three graphs present different information but are together.
Response 14: Figure 1A presents the yield of EPS in WT, AOX-overexpressed and AOX-silenced strains. Figure 1B shows that AOX gene influence the yield of EPS by genes PMI and PGM. And figure 1C shows that there was a significant positive correlation between EPS yield and ROS levels. Though the three graphs present different information, they all focus on how AOX genes affect polysaccharide production.
Comments 15: Line 271. It is not clear what the criteria for the selection of strains are. Relative expression of genes? Protein content?, Polysaccharide content?
Response 15: The criteria for the selection of strains is relative expression of AOX gene. The manuscript have been revised to enhance the clarity.
Comments 16: Lines 290-293. The sentence "The UV spectra of WE-N, WE-A, OE-N, OE-A, iE-N and iE-A showed no absorption peaks at 280 nm and 230 nm (Figure 4A), indicating that the sample contained almost no protein. This result is consistent with the protein content determination of the crude polysaccharide" is contrary to the previous sentence (lines 269-270) "The protein contents of crude polysaccharides from the WT, OE-AOX2 and AOXi25 strains were 1.62%, 3.21% and 10.73%, respectively, as determined using the BCA protein assay kit."
Response 16: The previous sentence in lines 269-270 describes the protein content of crude polysaccharides detected by BCA protein assay kit. While sentence in lines 290-293 describes the protein content of polysaccharides which is purified by using a DEAE-650M cellulose anion exchange chromatography column then detected by the UV (A) and FT-IR (B) spectra. So the protein content is different.
Comments 17: Lines 310-311. The sentence "A deeper examination of the infrared spectra reveals that polysaccharides share many similar characteristic absorption peaks within this spectral range." is incomplete because the discussion does not continue (not show graph or indicate the peaks).
Response 17: The sentence have been deleted.
Comments 18: Lines 323-328. Rewriting the sentence "Additionally, a relationship was observed between the polysaccharides structure and....... the physical structure of G. lucidum polysaccharides." to relate to the previous sentence or removing.
Response 18: The sentence "Additionally, a relationship was observed between the polysaccharides structure and....... the physical structure of G. lucidum polysaccharides." have been removed.
Comments 19: Lines 331-348. The sugars described in section 3.4.3 and Table 1 are neutrals. It would be important to discuss why they form acidic polysaccharides.
Response 19: We have included the discussion on why neutral monosaccharides can form acidic polysaccharides.
Comments 20: Line 355. Vitamin C was defined as Vc. Remove the words "vitamin C".
Response 20: VC is the abbreviation of vitamin C. We have uniformed the use of “VC”. The words "vitamin C" have been revised to VC.
Comments 21: Lines 351-368. If the acidic polysaccharides have high ABTS+ radical scavenging abilities, it is essential to discuss why, according to the results, they are composed only of neutral sugar.
Response 21: The acidic polysaccharides might be attributed to the presence of more uronic acid and sulfate. We have included the discussion.
Comments 22: Lines 369-385. In the section 3.5.2. (Hydroxyl radical scavenging ability), Why suggest that only AOX-silenced strains may contain higher uronic acid levels if both AOX-silenced and AOX-overexpression strains have higher antioxidant activity than the wild type?
Response 22: Figure 6B indicates that IE-A exhibits higher antioxidant activity than OE-A at a concentration of 0.5 mg/mL. As the concentration increases, the antioxidant activity of OE-A is enhanced; however, IE-A remains the most potent at 2.5 mg/mL. These results suggest that AOX-silenced strains may contain higher levels of uronic acid compared to AOX-overexpressing strains.
Reviewer 2 Report
Comments and Suggestions for Authors
Line 47: ... fungi, and protozoa, ... Indicate in Fungi: Which Phyla ... Basidiomycota?
Line 57: Has the AOX gene been observed in other Ganoderma species?
Line 63: How is production balanced between ganoderic acids and polysaccharides?
Lines 74-8: Avoid results in the introduction. Include hypothesis/research question
Lines 79-80: Perspectives in the introduction?
Line 196: Results and Discussion?
Line 250: Have PMI and PGM genes been observed in other Ganoderma species?
Lines 271-4: Avoid Materials and Methods in Results and Discussion
Lines 327-8: Is there evidence in other fungal species that the AOX gene and ROS level influence the physical structure of polysaccharides? Improve Discussion
Lines 336-7: Why do wild-type polysaccharides have higher molecular weights than genetically modified strains?
Author Response
Thank you very much for taking the time to review this manuscript. Please find the detailed responses below and the corresponding revisions/corrections highlighted/in track changes in the re-submitted files.
Comments 1: Line 47: ... fungi, and protozoa, ... Indicate in Fungi: Which Phyla ... Basidiomycota? Ascomycota、
Response 1: AOX is widely present in Basidiomycota and Ascomycota, including Ganoderma lucidum, Rhizoctonia oryzae, Aspergillus fumigatus, A. niger, Sclerotinia sclerotiorum, Paracoccidioides brasiliensis, Monascus ruber, C. albicans. It also appears in Zygomycota, but its role is less well studied in these groups.
Comments 2: Line 57: Has the AOX gene been observed in other Ganoderma species?
Response 2: Currently, there is a lack of research on the AOX gene in other Ganoderma species. We only find that AOX affects the biosynthesis of ganoderic acid through regulation of intracellular ROS levels in Ganoderma lucidum.
Comments 3: Line 63: How is production balanced between ganoderic acids and polysaccharides?
Response 3: In Ganoderma lucidum, the production of ganoderic acids and polysaccharides is typically influenced by various factors, including the growth conditions, genetic regulation, and metabolic pathways.
When carbon and nitrogen sources are abundant, Ganoderma lucidum tends to prioritize growth, which promotes the synthesis of polysaccharides. Conversely, when nutrients are limited, the fungus may shift its metabolic focus toward secondary metabolism, including the synthesis of ganoderic acids, to help it adapt to stress. There is also some cross-talk between the biosynthetic pathways for ganoderic acids and polysaccharides, and the production of one may influence the synthesis of the other under certain conditions. For example, when the organism is under nutrient stress, both the production of ganoderic acids and polysaccharides may be upregulated as part of a broader metabolic shift to enhance survival.
Stressors like temperature changes, oxygen availability, and pH fluctuations can influence the metabolic pathways. Favorable stressors promote the synthesis of ganoderic acids and polysaccharides.
Besides, ganoderic acids and polysaccharides are synthesized via different biochemical pathways. Ganoderic acids are produced through the terpenoid pathway, while polysaccharides are produced through carbohydrate metabolism pathways. Genetic regulation plays a crucial role in determining how these pathways are activated or suppressed.
In summary, the balance between ganoderic acids and polysaccharides in Ganoderma species is complex and is likely a result of both genetic factors and environmental influences. All of these aspects require in-depth research.
Comments 4: Lines 74-8: Avoid results in the introduction. Include hypothesis/research question
Lines 79-80: Perspectives in the introduction?
Response 4: we have rewrite in the manuscript.
Here is the details “This study investigated the impact of the AOX gene on the synthesis of Ganoderma lucidum polysaccharides, analyzing the underlying mechanisms with a focus on intracellular ROS levels and the key enzymes involved in polysaccharide biosynthesis. Additionally, the effects of the AOX gene on the physicochemical properties, structure, and biological ac-tivity of polysaccharides synthesized by engineered strains were examined.”
Comments 5: Line 196: Results and Discussion?
Response 5: The section have been modified.
Comments 6: Line 250: Have PMI and PGM genes been observed in other Ganoderma species?
Response 6: Currently, PMI and PGM genes are primarily observed in C. cinerea, C. militaris, G. frondosa, G. lucidum, and Hericium erinaceus. We didn’t find any article study PMI and PGM genes in other Ganoderma species.
Comments 7: Lines 271-4: Avoid Materials and Methods in Results and Discussion
Response 7: We have rewrite the lines 271-4 in the manuscript.
Comments 8: Lines 327-8: Is there evidence in other fungal species that the AOX gene and ROS level influence the physical structure of polysaccharides? Improve Discussion
Response 8: Although there is no direct evidence in other fungal species demonstrating that the AOX gene and ROS levels influence the physical structure of polysaccharides, numerous studies have shown that AOX plays a significant role in regulating intracellular ROS levels. In Aspergillus fumigatus, AOX confers resistance to oxidative stress. Similarly, in Aspergillus niger, AOX expression is involved in the response to heat shock, oxidative stress, and osmotic stress, potentially influencing carbohydrate metabolism and citric acid production. In Sclerotinia sclerotiorum, AOX regulates growth, development, and resistance to oxidative pressure. The reduction of ROS is crucial in fungal responses to various stress conditions. Therefore, the changes in the physical structure of polysaccharides in Ganoderma lucidum may serve as a protective mechanism against environmental stress.
Comments 9: Lines 336-7: Why do wild-type polysaccharides have higher molecular weights than genetically modified strains?
Response 9: AOX influence the monosaccharide composition of Ganoderma lucidum polysaccharides. Table 1 show that genetically modified strains had a higher proportion of low molecular weight monosaccharides. It may result in the polysaccharides molecular weights lower. Besides, the antioxidant activity of polysaccharides depends on position of the functional groups (e.g., hydroxyl, sulfate, amino, carboxyl, and phosphate groups). From our study, we can see genetically modified strains show stronger antioxidant activity. So the functional groups may also influence the molecular weight of polysaccharides.
Reviewer 3 Report
Comments and Suggestions for Authors
The paper "AOX Affects the Synthesis of Polysaccharides by Regulating the Reactive Oxygen Species in Ganoderma lucidum" present an interesting word. But it needs to be revisied.

Author Response
Thank you very much for taking the time to review this manuscript. Please find the detailed responses below and the corresponding revisions/corrections highlighted/in track changes in the re-submitted files.
Comments 1: - Line 22: replace “masses were 1.68 × 103” by “masses were 1.68 × 103”, with “3” as superscript.
- Throughout the text: replace “ABTS+” by “ABTS+”, with “+” as superscript.
- Line 26: replace “G. lucidum” by “Ganoderma lucidum”, since it is announced at the first time.
- Line 54: “Arabidopsis thaliana” must be in italic.
- Lines 56 and 57: “G. lucidum” must be in italic.
Response 1: All format questions have been modified.
Comments 2: - Throughout the text, why is AOX sometimes in italics and sometimes not? Same remark for the Supplementary document.
Response 2: The AOX gene is italicized when mentioned, whereas the AOX protein is not italicized.
Comments 3: - Line 88: replace “E.coli” by “E. coli”, i.e. it must be in italic and we must have a space between “E.” and “coli”.
- Lines 96 and 113: replace “table S1 in” by “Table S1 in”.
- Line 155: correct the following subtitle: “2.7.1. .ABTS+ radical scavenging activity”. i.e. delete
the extra “.” and the extra spaces between “2.7.1.” and “ABTS+”.
- Lines 158 and 163: please add the value of the room temperature by average± SD.
- Line 165: add “(1)” after “formula”.
- Lines 166, 177: delete the indent and replace “Where” by “where”.
- Line 171: replace “Briefly, In the” by “Briefly, in the”.
- Line 175: replace “510 nm, with VC” by “510 nm, with Vc”, since you used this abbreviation in line 162.
- Line 176: add “(2)” after “formula”.
- Line 180: correct the following subtitle: “2.7.3. .DPPH radical scavenging activity”. i.e. delete the extra “.” and the extra spaces between “2.7.3.” and “DPPH”.
- Line 182: replace “Shen et al” by “Shen et al.”.
- Line 189: delete the indent and add “where” at the beginning.
- Line 210: For the first reference to Figure 1, you used Figure 1B. Why not Figure 1A for the order logic?
- Line 230: replace “silenced strains. while their” by “silenced strains, while their”.
- Line 231: there are extra spaces between “strains,” and “except for SOD1”.
Response 3: All format questions have been modified.
Comments 4: - Add “Abbreviation section” to define all used abbreviations.
Response 4: we have added Abbreviation section
Comments 5: - Figures 1 and 2: add the significance of “*” used in the figures. The sentence talking about this should be added in the figures’ titles.
Response 5: The significance of “*” have been added in the figures’ titles.
Comments 6: - Materials and Methods section: please add the information about the number of repetitions carried out in your experiments. In fact, we have the errors bars in Figures without this information.
Response 6: All the Figures have the errors bars.
Comments 7: - Line 247: add the standard deviation for “3.74 g/L” by using “± SD”.
- Line 249: add the standard deviation for “0.58g/L and 0.53g/L” by using “± SD”.
Response 7: “± SD” has been added.
Comments 8: - Figures 2B: the standard deviation of the last histogram for PGM case corresponding to AOXi25, is high please repeat this experiment to check and to reduce the standard deviation.
Response 8: we have repeated this experiment and replaced the figure.
Comments 9:- Line 256: replace “(R2=0.77)” by “(R2=0.77)”, with 2 in exponent.
- Lines 362, 383, 384, 399 and 400: Please add the standard deviation to the all percentage value used.
- Line 281: replace “eluted first, This” by “eluted first. This”.
Response 9: All format questions have been modified.
Comments 10: - All Figures: the quality of all figure are very low, especially Figures 3, 4 and 5.
Response 10: The Figures have been replaced to enhance their quality.
Comments 11:- Table 1: please add one space between “Monosaccharide composition” and “(%)”.
Response 11: We have modified it
Comments 12:- Line 162, 175, 355, and 359: you used “Vc” or “VC” differently for “Ascorbic acid” and for “vitamin C”. Please check out this information. Moreover, please uniform the use of “Vc” or “VC” (you should use one from them) for the whole text and figures.
Response 12: We have uniformed the use of “VC”
Comments 13:- Lines 373 to 375: you said “As shown in Figure 6B, there was a positive correlation between the hydroxyl radical scavenging rate and the concentration of six polysaccharides.”. This interpretation is not clear sufficiently.
Response 13: We have rewritten the description.
Comments 14:- Lines 375-376: you used the sentence “the scavenging rates of acidic polysaccharides were significantly higher than those of neutral polysaccharides.”, have you carried out a statistical test here?
Response 14: We have removed that statement.
Comments 15:- Lines 387 to 404: the text is in Bold format, and it is not in Justified Style.
Response 15: We have modified the format of this paragraph.
Round 2
Reviewer 1 Report
Comments and Suggestions for Authors
A few minor suggestions are given.
- Line 216. The manuscript appears as Figure 1A, but in the figure foot, it is Figure 1.
- Line 287. Typing mistake. Acidic polysaccharides are adsorbed onto the exchanger, while neutral polysaccharides are not and are eluted first, This allows the separation of acidic and neutral polysaccharide components.
- Line 298. For clarity, the text ¨This result is consistent with the protein content determination of the crude polysaccharide¨ can be rewritten to ¨This result is consistent with the low protein content of the crude polysaccharide¨
- Lines 328-239: It is suggested that the text ¨A possible explanation is that AOX gene and ROS level influence the physical structure of G. lucidum polysaccharides¨ should be erased because it creates more doubts than explanations.
- Figure 8. Homogenize the text in the figure caption: WE-N or WT-N; WE-A or WT-A.
- Wild-type WT was defined as (WT) three times in lines 18, 89, and 213.
- Lines 392-393. For clarity, the text ¨These findings suggest that AOX-silenced strains may contain higher levels of uronic acid¨ can be rewritten to ¨These findings suggest that AOX-silenced strains may contain higher levels of uronic acid than AOX-overexpressed¨
Author Response
Thank you very much for taking the time to review this manuscript. Please find the detailed responses below and the corresponding revisions/corrections highlighted/in track changes in the re-submitted files.
Comments 1: Line 216. The manuscript appears as Figure 1A, but in the figure foot, it is Figure 1.
Line 287. Typing mistake. Acidic polysaccharides are adsorbed onto the exchanger, while neutral polysaccharides are not and are eluted first, This allows the separation of acidic and neutral polysaccharide components.
Response 1: We have revised the mistakes.
Comments 2: Line 298. For clarity, the text ¨This result is consistent with the protein content determination of the crude polysaccharide¨ can be rewritten to ¨This result is consistent with the low protein content of the crude polysaccharide¨
Response 2: We have rewritten the text.
Comments 3: Lines 328-239: It is suggested that the text ¨A possible explanation is that AOX gene and ROS level influence the physical structure of G. lucidum polysaccharides¨ should be erased because it creates more doubts than explanations.
Response 3: We have erased this part.
Comments 4: Figure 8. Homogenize the text in the figure caption: WE-N or WT-N; WE-A or WT-A.
Response 4: We have revised the mistakes.
Comments 5: Wild-type WT was defined as (WT) three times in lines 18, 89, and 213.
Response 5: We have modified lines 89 and 213 “wide typr (WT)” into “WT”.
Comments 6: Lines 392-393. For clarity, the text ¨These findings suggest that AOX-silenced strains may contain higher levels of uronic acid¨ can be rewritten to ¨These findings suggest that AOX-silenced strains may contain higher levels of uronic acid than AOX-overexpressed¨
Response 6: We have rewritten this text.